# A randomized trial evaluating virus-specific effects of a combination probiotic in children with acute gastroenteritis

Stephen B. Freedman [1✉], Jianling Xie[1], Alberto Nettel-Aguirre[1], Xiao-Li Pang[2], Linda Chui[2], Sarah Williamson-Urquhart[1], David Schnadower[3], Suzanne Schuh[4], Philip M. Sherman [4], Bonita E. Lee[2], Serge Gouin[5], Ken J. Farion [6], Naveen Poonai[7], Katrina F. Hurley[8], Yuanyuan Qiu[2], Binal Ghandi[2], Colin Lloyd[2] & Yaron Finkelstein[4], the Pediatric Emergency Research Canada Probiotic (PERC) Regimen for Outpatient Gastroenteritis Utility of Treatment (PROGUT) Trial Group*

Gastroenteritis accounts for nearly 500,000 deaths in children younger than 5 years annually. Although probiotics have been touted as having the potential to expedite diarrhea resolution, recent clinical trials question their effectiveness. A potential explanation is a shift in pathogens following the introduction of a rotavirus vaccine. Here, we report the results of a multi-center, double-blind trial of 816 children with acute gastroenteritis who completed follow-up and provided multiple stool specimens. Participants were randomized to receive a probiotic containing *Lactobacillus rhamnosus* and *Lactobacillus helveticus* or placebo. We report no virus-specific beneficial effects attributable to the probiotic, either in reducing clinical symptoms or viral nucleic acid clearance from stool specimens collected up to 28 days following enrollment. We provide pathophysiological and microbiologic evidence to support the clinical findings and conclude that our data do not support routine probiotic administration to children with acute gastroenteritis, regardless of the infecting virus.

[1] Alberta Children's Hospital Foundation Professor in Child Health and Wellness, Alberta Children's Hospital Research Institute, Cumming School of Medicine, University of Calgary, 28 Oki Drive NW, Calgary, AB T3B 6A8, Canada. [2] University of Alberta, 116 St & 85 Ave., Edmonton, AB T6G 2R3, Canada. [3] University of Cincinnati, 3333 Burnet Ave, Cincinnati, OH UCA 45229, USA. [4] University of Toronto, 555 University Avenue, Toronto, ON M5G 1X8, Canada. [5] Université de Montréal, 3175 Chemin de la Côte-Sainte-Catherine, Montréal, QC H3T 1C5, Canada. [6] University of Ottawa, 401 Smyth Rd, Ottawa, ON K1H 8L1, Canada. [7] University of Western Ontario, 800 Commissioners Road E, London, ON N6A 5W9, Canada. [8] Dalhousie University, 5980 University Avenue, PO Box 9700, Halifax, NS B3K 6R8, Canada. *A list of authors and their affiliations appears at the end of the paper. ✉email: Stephen.freedman@ahs.ca

It is estimated that 0.57 acute gastroenteritis (AGE) episodes per person-year occur in Canada, amounting to nearly 19.5 million episodes annually[1], while in the United States, over 48 million episodes occur each year[2,3]. Despite the availability of a vaccine against rotavirus[4], which has led to precipitous decreases in hospitalizations[5] and emergency department (ED) visits[6] attributable to rotavirus gastroenteritis, coverage remains far from universal; on the global scale, rotavirus remains the leading etiology of diarrhea-associated mortality[7]. Other viruses also contribute significantly to the burden of disease in North America, where norovirus now represents the leading etiology of medically-attended AGE[8,9]. Because there are no widely accepted effective treatment options available beyond supportive care, health care providers and affected individuals continue to explore a variety of options, including probiotics[10].

Although the administration of probiotic agents to children with AGE and diarrhea has been recommended by international clinical guidelines[11–13], we recently conducted one of the largest randomized clinical trials to date, and found no benefits associated with probiotic administration. The trial, which included 886 children with AGE (816 of whom completed provided a stool specimen and completed follow-up), reported that those who received a 5-day course of *L. helveticus/L. rhamnosus* did not have significantly different odds of experiencing moderate-to-severe AGE following randomization compared with those administered a placebo [OR: 1.06 (95% CI, 0.77 to 1.46)][14]. These findings were supported by a simultaneously conducted trial in the United States, which evaluated a different probiotic product, containing *L. rhamnosus* GG[15]. A potential explanation for the lack of benefit associated with probiotic administration in these studies is that the benefits may be pathogen specific (e.g., beneficial in rotavirus infection but not norovirus)[16]. This explanation is supported by the diverse underlying pathophysiologic processes induced by different etiologic pathogens[17] and the multiple proposed mechanisms of action of probiotics[18].

Understanding pathogen-specific effects is increasingly important since rotavirus vaccination programs have substantially altered the target pathogen population. Advances in molecular diagnostics have also enabled the identification of enteropathogens in more than 75% of stool specimens submitted by children with AGE[19], revealing a shift from rotavirus to norovirus as the most common identified pathogen among individuals with AGE seeking medical care in the United States[8,20]. Moreover, real-time reverse transcription-polymerase chain reaction testing has identified a wide range of viral loads in norovirus-associated AGE cases[21] with high loads being associated with more severe symptoms[22–24], prolonged hospitalization and viremia[23,25]. After attaining a peak level during an AGE episode, stool viral loads decrease in a time-dependent manner; the higher the initial viral load, the longer the time required for clearance from stool[26]. As such, the ability to reduce the intestinal viral load more rapidly would represent an objective, pathogen-specific method of how probiotics could modulate AGE infections.

The aforementioned shift in etiologic pathogens, the ability to identify pathogens in real-time[27], the huge market share and concerns regarding money spent on probiotics[28] and the recent publication of two studies that question the benefits commonly touted for probiotics[14,15], highlight the importance of understanding the pathophysiologic pathogen-specific potential benefits of probiotic administration. Because the aforementioned clinical trials relied on caregiver report of clinical symptoms, they did not explore potential therapeutic effects at individual patient and pathogen levels.

To address this point, as an integral part of the aforementioned placebo-controlled, randomized, parallel-arm, clinical trial[14], we identify pathogens in collected stool specimens to enable an evaluation of the ability of an orally administered probiotic to reduce symptom severity at a pathogen-specific level. We also assess the changes, from baseline, in viral loads in stool specimens at the end of the probiotic course (on day 5 of treatment) and 4 weeks after randomization (on day 28), relative to placebo. We determine that the probiotic has no pathogen-specific beneficial effects compared to placebo, either in reducing clinical symptoms or clearance of viral nucleic acid from stool specimens collected up to 28 days following enrollment. We provide pathophysiological and microbiologic evidence to support the clinical findings, and conclude that our data do not support routine probiotic administration to children with AGE.

## Results

**Participants.** Of the 886 children who were enrolled into the clinical trial between November 5, 2013, and April 7, 2017, 816 (92.1%) provided a baseline stool specimen and completed follow-up; Fig. 1. Demographic and clinical characteristics of the probiotic and placebo groups are summarized in Table 1; there were no meaningful differences in index visit clinical parameters. A virus was only detected in 451 (55.3%) children, bacteria only in 37 (4.5%), virus/bacteria co-detection in 19 (2.3%), parasite-viral/parasite co-detection in 10 (1.2%), and no pathogen was identified in 299 (36.6%) participants; Supplementary Table 2. There were differences noted regarding the distribution of pathogens among patient groups with respect to age, rotavirus vaccination status, presence of vomiting, and number of diarrhea episodes; supplementary Table 3.

**Primary outcome.** No differences were detected in the mean post-randomization modified Vesikari scale (MVS) scores between probiotic and placebo groups for any of the five categories of pathogens analyzed; Table 2 and Fig. 2. The weighted linear regression model fitted with interaction terms and covariates revealed no significant associations between post-randomization MVS scores and treatment allocation; supplementary Table 4. The interaction terms were removed from the regression model as they were not significant; after removal, the findings were unchanged; supplementary Table 5. Statistically significant associations with the post-randomization MVS score were identified for age (−0.054; 95%CI: −0.083, −0.025 per 1month increase) and baseline MVS score (0.339; 95%CI: 0.198, 0.480 per 1 point increase). In our linear regression models that included a priori identified covariates and only individual pathogen groups and pathogens, no significant association between the post-randomization MVS score and treatment allocation were identified for the following groups: test negative, isolated bacteria, isolated virus, virus/bacteria co-detection, parasite and parasite/virus co-detection, adenovirus, norovirus, rotavirus, and *Campylobacter* spp.; Supplementary Tables 6–17.

**Secondary outcomes.** There were no clinically significant differences between the 148 children who provided all three stool samples and those who provided only one or two samples; Supplementary Table 18. An insufficient number of participants had bacterial infections to enable a statistically robust analysis of bacterial pathogen load reduction. Comparing the pathogen load reductions among participants with adenovirus, norovirus, or rotavirus, there were no significant differences between participants administered probiotic or placebo between days 0 to 5, and 5 to 28; Table 3. Probiotic administration was not associated with pathogen load reduction in any of the 9 linear regression models constructed for the 3 viruses (i.e., adenovirus, norovirus, rotavirus) and 2 time intervals (i.e., 0 to 5 or 5 to 28 days).

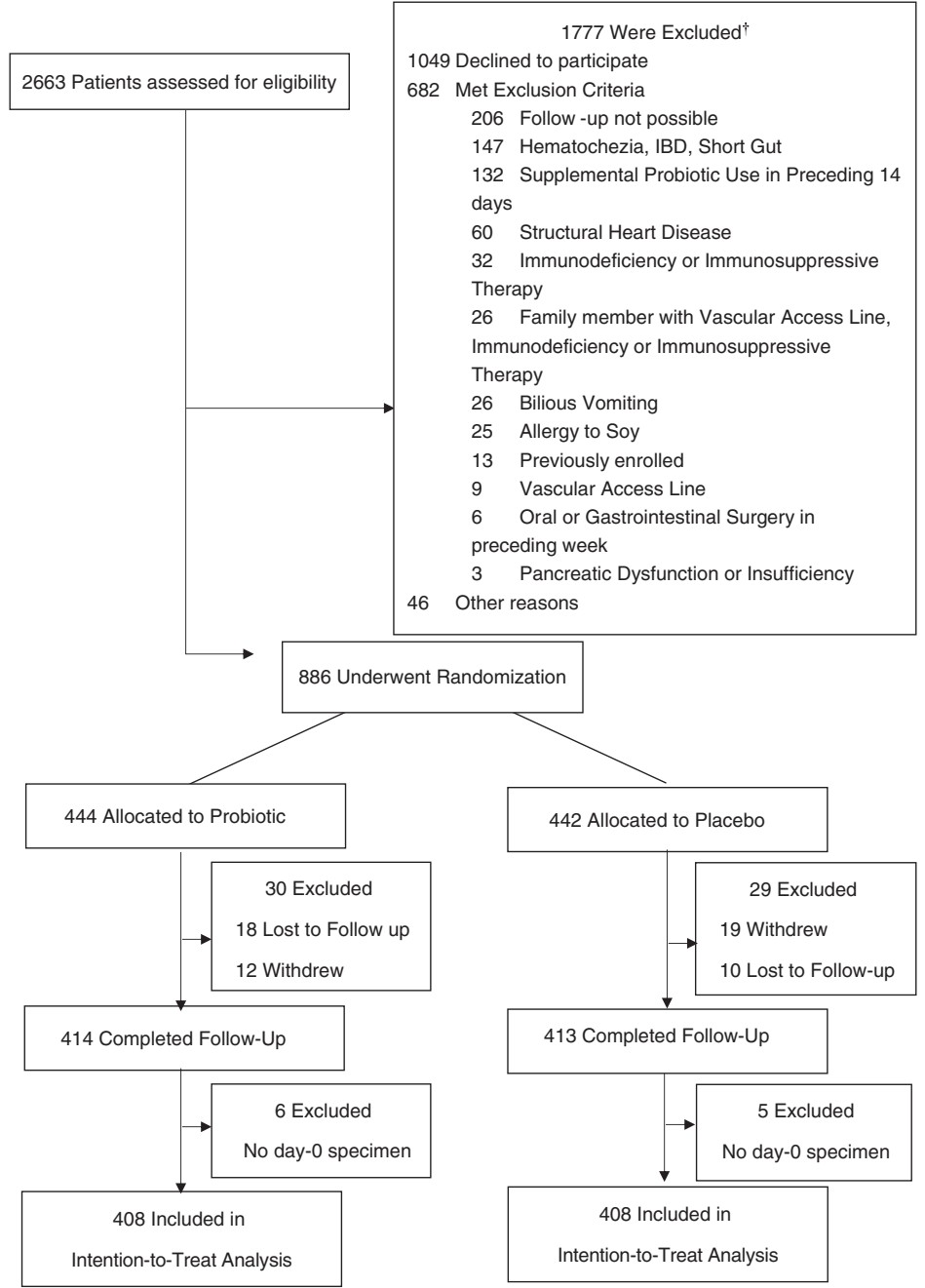

**Fig. 1 Flow diagram of patient cohort.** IBD inflammatory bowel disease; †Patients may have met more than one criterion.

We found associations between the Day 0 pathogen (per 1.0 increase in Log10 nucleic acid/gm stool) load and mean reductions of adenovirus pathogen loads between Day 0 to 5 [0.33 Log10 nucleic acid/gm stool (95%CI: 0.17, 0.49)]. Similarly, we found associations between Day 0 rotavirus pathogen load and the Day 0 to 5 reduction [1.05 Log10 nucleic acid/gm stool (95% CI: 0.84, 1.25)]. However, for norovirus, no significant associations were identified. Older children experienced a greater mean norovirus pathogen load reduction [Day 5 to 28, 0.08 Log10 nucleic acid/g stool (95%CI: 0.02, 0.15) per 1 month increase] compared to their younger counterparts.

Although the raw baseline MVS score was associated with the rotavirus Log10 transformed stool Day 0 viral load ($P = 0.05$) in the linear regression model, the associations for norovirus ($P = 0.11$) and adenovirus ($P = 0.75$) were not significant. Similarly,

only the Day 5 rotavirus Log10 transformed stool viral load was significantly associated with the follow-up MVS score ($P = 0.03$). The overall similarities of the declines in the Log10 transformed pathogen loads between the probiotic and placebo groups are displayed in Figs. 3–5.

**Exploratory outcomes.** Sub-group analyses were conducted based on participant age and pathogen detected. In these models, the interaction between treatment assignment and age was not statistically significant when the MVS score was set as the dependent variable and the models were adjusted for other relevant covariates. Similarly, we analyzed the primary outcome, MVS score, based on breast-feeding status and found no evidence of interaction with treatment assignment; Supplementary Table 20. Adverse events did not differ between groups, as previously reported[14].

**Table 1 Clinical characteristics by treatment groups.**

| Characteristics | Probiotics $N = 408$ | Placebo $N = 408$ | P- value[††] |
|---|---|---|---|
| Age (month), median (IQR) | 15.0 (10.0, 24.5) | 16.0 (10.0, 24.0) | 0.680 |
| Male sex (no., %) | 229 (56.1) | 235 (57.6) | 0.724 |
| Weight median (IQR) (kg) | 10.5 (9.0, 13.0) | 10.7 (8.9, 12.6) | 0.911 |
| Exclusive breast fed (no., %) | 22 (5.4) | 29 (7.1) | 0.386 |
| Received antibiotics in previous 14 days (no., %) | 50 (12.3) | 58 (14.2) | 0.470 |
| Received rotavirus vaccine (no., %) | | | 0.827 |
| Yes | 195 (47.8) | 196 (48.0) | |
| No | 116 (28.4) | 109 (26.7) | |
| Unsure | 97 (23.8) | 103 (25.2) | |
| Duration of illness mean (SD) (h)[†] | 43.3 (22.9) | 43.2 (20.0) | 0.929 |
| Baseline modified Vesikari scale score–mean (SD)[‡] | 11.2 (2.7) | 10.9 (2.8) | 0.161 |
| Vomiting (no., %) | 322 (78.9) | 302 (74.0) | 0.117 |
| No. of vomiting episodes in preceding 24 h–median (IQR)[§] | 4 (2, 6) | 4 (2, 7) | 0.319 |
| No. of diarrhea episodes in preceding 24 h–median (IQR) | 5 (3, 8) | 5 (3, 8) | 0.180 |
| Febrile—no. (%)[¶] | 182 (44.6) | 179 (43.9) | 0.888 |
| Clinical dehydration scale score–median (IQR)[‖] | 1 (0, 2) | 0 (0, 2) | 0.139 |
| Received ondansetron at index visit—no. (%) | 90 (22.1) | 89 (21.8) | >0.99 |
| Received antibiotics at index visit/recommended at discharge—no. (%) | 11 (2.7) | 4 (1.0) | 0.115 |
| Received intravenous rehydration at index visit—no. (%) | 36 (8.8) | 31 (7.6) | 0.610 |
| Admitted to hospital at index visit—no. (%) | 10 (2.5) | 10 (2.5) | >0.99 |

IQR interquartile range. SD standard deviation, no. number.
[†]This variable was defined according to the duration of vomiting or the duration of diarrhea before enrollment, whichever was greater.
[‡]Scores on the modified Vesikari scale range from 0 to 20, with higher scores indicating greater disease severity.
[§]The denominator for this variable was the number of children who had vomiting.
[¶]Febrile was defined as a documented adjusted rectal temperature of at least 38.0 °C.
[‖]Scores on the clinical dehydration scale range from 0 to 8, with higher scores indicating more severe dehydration.
[††]Statistical tests performed included the T-Test and Mann–Whitney U Test for means and medians, respectively, and the Chi-square test for categorical variables. P-values reported are two-sided and unadjusted for multiple comparison. A P-value <0.0029 is statistically significant for comparisons included in this table after adjustment for multiple comparison using the Bonferroni method ($n = 17$).

**Table 2 Primary outcome–modified Vesikari scale score by treatment and pathogen groups.**

| | Overall $N = 816$ Mean (SD) | Probiotics $N = 408$ N; Mean (SD) | Placebo $N = 408$ N; Mean (SD) | Mean difference (95%CI) | P-value[†] |
|---|---|---|---|---|---|
| Negative ($N = 299$) | 5.3 (4.2) | 139; 5.1 (4.1) | 160; 5.5 (4.3) | −0.410 (−1.379, 0.560) | 0.407 |
| Virus only ($N = 451$) | 6.1 (4.6) | 232; 6.3 (4.7) | 219; 6.0 (4.4) | 0.263 (−0.589, 1.114) | 0.545 |
| Bacteria only ($N = 37$) | 7.7 (4.9) | 17; 9.4 (5.3) | 20; 6.4 (4.2) | 2.990 (−0.160, 6.142) | 0.063 |
| Virus/bacteria co-detection ($N = 19$) | 6.3 (4.8) | 13; 6.2 (4.4) | 6; 6.4 (5.9) | −0.180 (−5.544, 5.184) | 0.948 |
| Parasite ($N = 6$) or parasite/ virus co-detection ($N = 4$) | 6.7 (5.1) | 7; 7.9 (5.6) | 3; 3.8 (1.4) | 4.038 (−0.730, 8.806) | 0.097 |

CI confidence interval, SD standard deviation.
[†]Calculated employing the T-Test. The P-values reported are two-sided and unadjusted for multiple comparison. A P-value < 0.001 is statistically significant after adjustment for multiple comparison using Bonferroni method ($n = 5$).

## Discussion

This study expands our knowledge by conducting several unique and novel virus-specific evaluations of the effectiveness of a combination probiotic in children with AGE. We found no indication that probiotic administration lessens the burden of disease, quantified by the MVS score, regardless of the etiologic pathogen group (i.e., virus, bacteria or parasite) or specific viral etiologies (i.e., adenovirus, norovirus or rotavirus). In addition, we found no evidence that children administered the probiotic agent experienced a more rapid clearance of pathogen from stool, compared to those administered placebo, either during the treatment course or over the subsequent weeks.

Earlier reports suggested that there may be pathogen-specific benefits associated with probiotic use[29], with the greatest benefits seen in children with rotavirus diarrhea and limited benefit in children infected with bacterial pathogens[30]. Following the routine administration of a rotavirus vaccine[31], norovirus has replaced rotavirus as the most common pathogen[8]. This pathogen shift may in part explain the lack of probiotics benefit in viral AGE in our study. While earlier studies and meta-analyses focused on evaluating the benefit of probiotics in the context of rotavirus infection, none have been sufficiently large to analyze groups of pathogens or specific viral pathogens other than rotavirus[32]. In this report, we clarify that no pathogen subgroups or specific viral subgroups were found to benefit from probiotic administration. Adjustment for the duration of illness at the time of the initiation of probiotic therapy[33] did not alter our findings.

Our stool pathogen load analysis constitutes a unique approach to evaluating the effect of probiotic therapy and is based on evidence that a higher stool viral load is associated with more severe disease and reflects a greater degree of intestinal epithelium damage[22,23,34]. The latter scenario can lead to viral spread beyond the intestines into the bloodstream[35,36]. Although we hypothesized that probiotic administration may more rapidly reduce the stool pathogen load, we did not identify significant associations to support this notion.

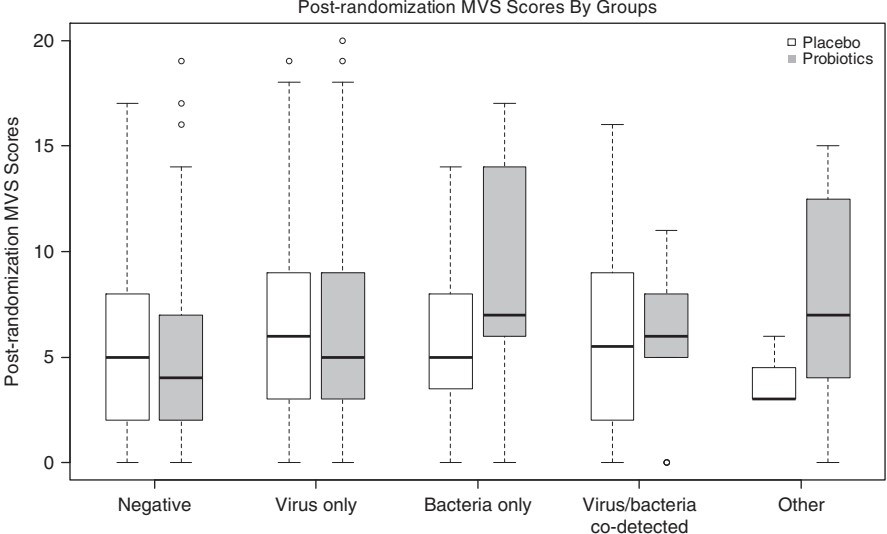

**Fig. 2 Post randomization disease severity (Modified Vesikari Scale scores) based on pathogen identified, compared between probiotic and placebo groups.** Boxplots of post-randomization modified Vesikari scale cores by treatment and pathogen groups (Tested negative: placebo $n = 160$, probiotics $n = 139$; virus only: placebo $n = 219$, probiotics $n = 232$; bacteria only: placebo $n = 20$, probiotics $n = 17$; virus/bacteria co-detection: placebo $n = 6$; probiotics $n = 13$; Other type detected including parasite only and virus/parasite co-detection: placebo $n = 3$, probiotics $n = 7$). In the box plot, the line that divides the box into two parts represents the median of the data; the upper and lower bounds of the box represent the 25% and 75 percentiles, respectively. The lower whiskers represent the 25%ile-(1.5× IQR) and the upper whiskers represent the 75%ile-(1.5× IQR). The dots beyond the whisker represents outliers of values within the dataset. Source data are provided as a Source Data file.

**Table 3 Stool pathogen reduction, follow-up stool specimens, probiotic versus placebo, log10 transformed copies/gm stool\*.**

| | Probiotics N; Log10 reduction | Placebo N; Log10 reduction | Difference (95%CI)‡ | P-value† |
|---|---|---|---|---|
| **Day 0–Day 5** | | | | |
| Adenovirus, mean (SD) | 25; 3.10 (2.27) | 19; 2.64 (2.02) | 0.46 (−0.87, 1.79) | 0.489 |
| Adenovirus, median (IQR) | 25; 3.25 (1.22, 5.09) | 19; 2.59 (0.98, 4.43) | 0.45 (−0.98, 1.85,) | 0.462 |
| Norovirus, mean (SD) | 29; 0.49 (1.44) | 43; 0.80 (1.86) | −0.31 (−1.13, 0.50) | 0.445 |
| Norovirus, median (IQR) | 29; 0.40 (−0.53, 0.98) | 43; 0.47 (0.18, 1.05) | −0.24 (−0.76, 0.21) | 0.349 |
| Rotavirus, mean (SD) | 28; 2.79 (2.60) | 21; 1.23 (5.02) | 1.56 (−0.90, 4.02) | 0.206 |
| Rotavirus, median (IQR) | 28; 2.77 (1.11, 4.79) | 21; 2.96 (0.41, 4.25) | 0.49 (−1.08, 2.11) | 0.423 |
| **Day 0–Day 28** | | | | |
| Adenovirus, mean (SD) | 14; 10.12 (3.06) | 8; 10.39 (2.37) | −0.27 (−2.90, 2.35) | 0.831 |
| Adenovirus, median (IQR) | 14; 11.54 (6.04, 12.53) | 8; 11.40 (8.08, 12.04) | −0.042 (−2.00, 2.00,) | 0.973 |
| Norovirus, mean (SD) | 18; 5.79 (2.39) | 24; 5.57 (2.35) | 0.22 (−1.27, 1.71) | 0.768 |
| Norovirus, median (IQR) | 18; 4.91 (4.10, 8.97) | 24; 4.90 (3.54, 7.14) | 0.44 (−1.31, 1.73) | 0.576 |
| Rotavirus, mean (SD) | 12; 9.92 (2.30) | 6; 6.02 (7.22) | 3.89 (−3.66, 11.45) | 0.249 |
| Rotavirus, median (IQR) | 12; 9.30 (7.87, 12.28) | 6; 8.71 (−1.40, 12.00) | 1.23 (−1.59, 12.22) | 0.616 |
| **Day 5–Day 28** | | | | |
| Adenovirus, mean (SD) | 14; 7.19 (2.22) | 8; 6.80 (1.86) | 0.38 (−1.55, 2.32) | 0.684 |
| Adenovirus, median (IQR) | 14; 7.72 (5.45, 8.40) | 8; 6.63 (5.14, 8.55) | 0.23 (−1.37, 2.50) | 0.868 |
| Norovirus, mean (SD) | 23; 5.62 (2.36) | 26; 4.83 (2.40) | 0.79 (−0.58, 2.16) | 0.252 |
| Norovirus, median (IQR) | 23; 4.74 (4.08, 7.63) | 26; 4.30 (2.81, 6.63) | 0.84 (−0.61, 1.94) | 0.167 |
| Rotavirus, mean (SD) | 13; 6.79 (3.37) | 8; 6.64 (2.77) | 0.16 (−2.82, 3.13) | 0.914 |
| Rotavirus, median (IQR) | 13; 5.86 (4.20, 10.32) | 8; 5.95 (4.71, 8.52) | −0.157 (−2.48, 3.52) | 0.916 |

*SD* standard deviation, *IQR* Interquartile range.
*All mean and median values represent Log10 transformed copies of gram per stool reduction (i.e., Day 0 value less the Day 5 value).
†Statistical significance assessed using Student's *T*-Test and Mann–Whitney *U* Test for means and medians, respectively. The *P*-values reported are two-sided and unadjusted for multiple comparison. A *P*-value <0.0028 is statistically significant after adjustment for multiple comparison using Bonferroni method ($n = 18$).
‡Hodges–Lehmann estimate test to calculate median difference.

Our study has several limitations. Only 18% of participants provided specimens at all three study time points. Consequently, we had fewer participants than was anticipated and thus several secondary analyses were potentially underpowered. While participants who submitted all three specimens could have differed from those who did not, there is no reason to assume a related systematic bias. Indeed, Supplementary Table 18 demonstrates that the two groups of children were clinically similar. However,

because of the small number of participants with bacterial and parasitic infections, we were unable to conduct robust pathogen load analyses related to bacteria and parasite clearance.

In conclusion, we observed no beneficial virus-specific clinical effects associated with the administration of a 5-day course of a *L. helveticus/L. rhamnosus* combination probiotic, for children with AGE. Similarly, probiotic administration did not result in more rapid clearance of viral pathogens from stool specimens,

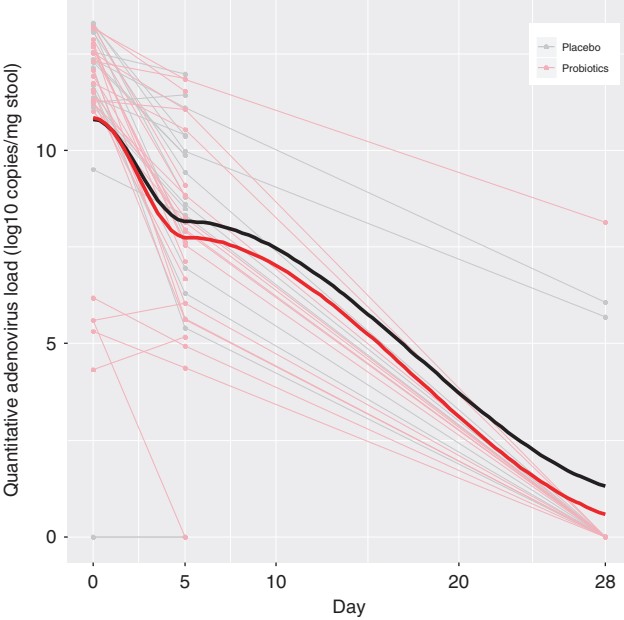

**Fig. 3 Change in adenovirus viral load quantity in stool over time, compared between probiotic and placebo groups.** Stool Log10 transformed copies of adenovirus by treatment group [probiotic ($N = 25$) vs. placebo ($N = 20$)] across time (in days) following randomization on the x-axis. Thin light gray lines refer to patients provided placebo, this red lines refer to those provided probiotic; thick black and red lines to locally weighted smoothing lines respectively. The two-sided P-value represents the result of a linear mixed effect model with random intercepts (subjects random effect) comparing placebo (reference group) vs. probiotic on viral load including time and treatment group variables, and an interaction term for the latter two. Source data are provided as a Source Data file. Adenovirus (mean difference: −0.12; 95% CI: −1.94, 1.70; $P = 0.90$).

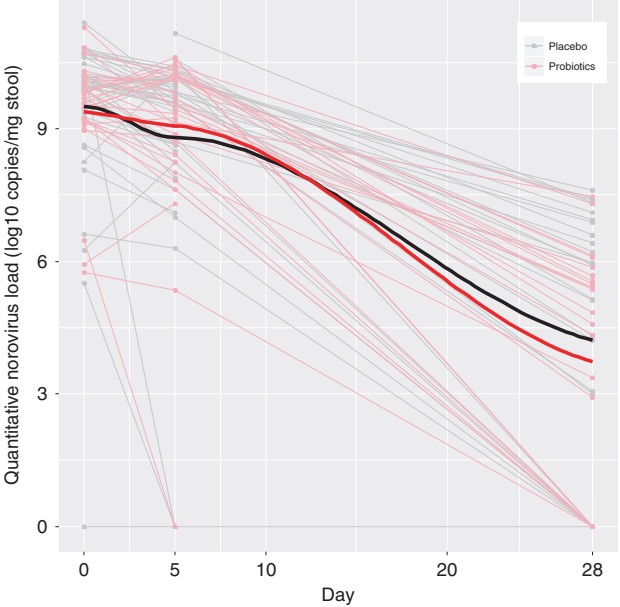

**Fig. 4 Change in norovirus viral load quantity in stool over time, compared between probiotic and placebo groups.** Stool Log10 transformed copies of norovirus by treatment group [probiotic ($N = 34$) vs. placebo ($N = 47$)] across time (in days) following randomization on the x-axis. Thin light gray lines refer to patients provided placebo, this red lines refer to those provided probiotic; thick black and red lines to locally weighted smoothing lines respectively The two-sided P-value represents the result of a linear mixed effect model with random intercepts (subjects random effect) comparing placebo (reference group) vs. probiotic on viral load, including time and treatment group variables, and an interaction term for the latter two. Source data are provided as a Source Data file. Norovirus (mean difference: 0.16; 95% CI: −0.81, 1.14; $P = 0.74$).

compared with placebo. These findings strengthen the conclusion that in children who present to an ED with viral-induced AGE, twice-daily administration of a combined *L. rhamnosus/L. helveticus* probiotic does not reduce the severity of AGE, or expedite the clearance of viruses in stool.

## Methods
**Study design and oversight**. The clinical trial design and methods have been published and the trial has been registered at www.clinicaltrials.gov: NCT01853124[37]. The protocol was approved by the research ethics boards at each of the six participating Canadian tertiary care pediatric centers located in Calgary (Conjoint Health Research Ethics Board), London (Western University Health Sciences Research Ethics Board), Toronto (SickKids Research Ethics Board), Ottawa (Children's Hospital of Eastern Ontario Research Ethics Board), Montreal (Comité d'éthique de la recherche du CHU Sainte-Justine) and Halifax (IWK Research Ethics Board), in Canada; the caregivers of all participants provided written informed consent. In brief, in this double-blind, placebo controlled trial, patients aged 3 to 48 months with AGE presenting for ED care, were randomly assigned, in a 1:1 ratio, to receive $4.0 \times 10^9$ colony forming units of a *L. rhamnosus* R0011 and *L. helveticus* R0052 (95:5 ratio) probiotic preparation or matching placebo twice daily for 5 days, in addition to usual care. Eligible children had ≥3 episodes of diarrhea in a 24-h period, and had vomiting or diarrhea for <72 h. All children were evaluated by a physician who assigned a diagnosis of AGE. Children were excluded if they or a person living in their household had a central venous line, structural heart disease, were immunocompromised, or were receiving immunosuppressive therapy. Children who presented with a history of oral or gastrointestinal surgery within the preceding 7 days, blood in their vomit or stool, bilious vomiting, a chronic intestinal disorder, pancreatic insufficiency, probiotic use in the preceding 14 days, soy allergy, and an inability to complete follow-up were also excluded.

A total of 816 participants completed 14 day follow-up and provided symptom outcome data. Attempts were made to collect stool specimens from all participants on Day 0 (ED enrollment), Day 5 (last day of probiotic/placebo administration), and Day 28 after enrollment. Only participants who tested positive for an

enteropathogen and provided stool specimens at multiple time points, were included in the current study.

**Objectives**. The primary objective was to determine if a 5-day probiotic treatment course administered to children with AGE resulted in pathogen-specific clinical benefits quantified using the validated and widely-used[38] MVS score[39,40]. Secondary objectives identified a priori included (1) assessing if probiotic administration resulted in a greater reduction in stool pathogen load compared with placebo; and determining the relationship between (2) correlating baseline (Day 0) stool pathogen load and baseline MVS score, and (3) Day 5 stool pathogen load and the follow-up MVS score.

**Modified Vesikari scale score**. MVS scores range from 0 to 20, with higher scores indicating more severe disease[39,40]. The MVS score quantifies severity over a broad range of symptoms and interventions among outpatients. This measurement tool was validated in two prospective cohort studies in similar patient populations[39,40] and has been employed in several clinical trials[14,15,41]. Baseline symptoms that occurred prior to the index ED visit were not included in the follow-up MVS score calculation.

**Randomization**. To sequentially assign children to probiotic or placebo, we employed a random-number–generating software, accessed through www.randomize.net, which was programed to use block sizes of 4 and 6, stratified according to site. The random allocation sequence was generated by the research pharmacy at the coordinating center. Participants were enrolled by research nurses or assistants at each site who provided caregiver with the allocation assignment. Participants and their parents or guardians, trial and clinical staff, and specimen and data analysts remained blinded to treatment assignment through the use of a placebo that was identical in appearance, smell, and weight to the intervention agent (i.e., probiotic).

**Specimen collection**. We attempted to collect a stool sample from all participants prior to ED discharge. If a specimen was not provided prior to ED discharge, caregivers were instructed to collect a stool sample at home, which was retrieved by a study-funded courier service.

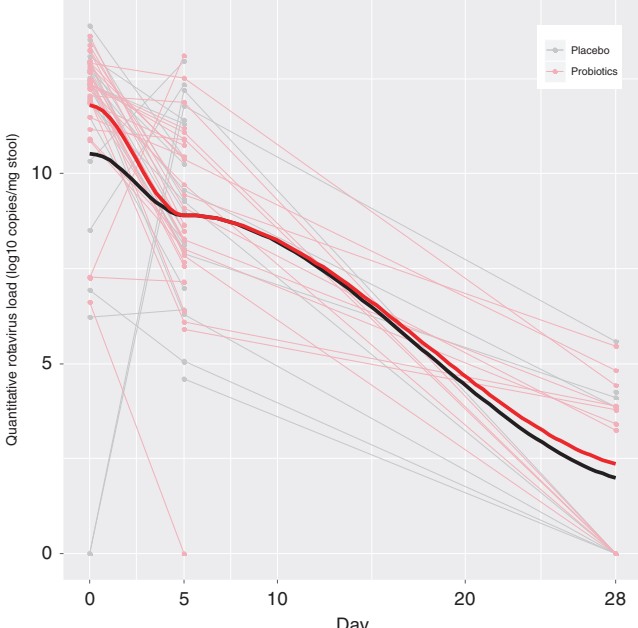

**Fig. 5 Change in rotavirus viral load quantity in stool over time, compared between probiotic and placebo groups.** Stool Log10 transformed copies of rotavirus by treatment group [probiotic ($N = 30$) vs. placebo ($N = 24$)] across time (in days) following randomization on the $x$-axis. Thin light gray lines refer to patients provided placebo, this red lines refer to those provided probiotic; thick black and red lines to locally weighted smoothing lines respectively. The two-sided $P$-value represents the result of a linear mixed effect model with random intercepts (subjects random effect) comparing placebo (reference group) vs. probiotic on viral load, including time and treatment group variables, and an interaction term for the latter two. Source data are provided as a Source Data file. Rotavirus (mean difference: 0.74; 95% CI: $-0.45$, 1.93; $P = 0.22$).

A rectal swab was performed with a flocked tipped sterile swab (FLOQSwabs™ Flocked Swabs, Copan) prior to ED discharge on all children not providing a stool specimen[19]. This approach allowed us to obtain a specimen for molecular pathogen identification prior to probiotic administration from all study participants. The swab was only tested if an ED stool specimen was not obtained.

Day 5 and Day 28 stool samples were requested from all study participants who provided a Day 0 stool sample, either while in the ED or at home. Caregivers were provided with collection instructions along with containers. Specimens were labeled with the date and time of collection and the subject's study identification number. They were returned to the research team by a study-funded courier service within 12 h of collection. All specimens were placed in coolers with ice packs while in transit to the laboratory. Upon receipt, each sample was split and frozen for future testing[42–44]. Sites then batch-shipped all frozen stool samples to the Alberta Public Laboratories-ProvLab (Edmonton, Alberta, Canada) bi-annually to enable interim laboratory analyses to verify collection and processing procedures. All stool tests were conducted blinded to treatment allocation and clinical symptoms.

**Pathogen detection.** All Day 0 specimens obtained in the ED underwent bacterial culture locally. A multiplex nucleic acid panel that detects 15 pathogens: enteric viruses, bacteria and parasites (Luminex xTAG Gastrointestinal Pathogen Panel) was later performed at the Provincial Laboratory for Public Health–Alberta Public Laboratorie-ProvLab[45]. To ensure that negative rectal swab test results were not due to insufficient stool obtained on the rectal swab, all patients with a Day 0 ED rectal swab that tested negative for all enteropathogens had testing repeated using the Day 0 stool specimen collected at home. Day 5 and 28 specimens were tested only if the Day 0 specimen tested positive for an enteropathogen.

**Pathogen load quantification.** Quantification procedures were standardized to ensure that the homogeneity and proportion of stool included in each analysis was consistent between samples (intra-patient and inter-patient) and hence per reporting unit (g). To achieve this degree of standardization, a 10% (weight/volume) suspension of stool specimen was prepared with phosphate-buffered saline (PBS) and clarified by centrifugation. Testing of the Day 0, 5, and 28 specimens from the same patient were performed in the same test run to eliminate inter-run variation.

Viral loads were quantified as previously described[46]. In brief, samples were thawed, then mixed by vortexing to prepare a 10% stool specimen suspension after centrifugation. Total nucleic acid was then extracted and eluted using the MagaZorb® total RNA Prep kit (Promega, Madison, WI). Nucleic acid extracted from non-study stool samples testing positive for well-characterized enteric viruses (i.e., adenovirus 40/41, norovirus, and rotavirus) were used as positive controls. The primers and probes for the detection of adenovirus, norovirus, and rotavirus[47–51] were labeled with Fam detector and Tamara quencher dyes (Applied Biosystems). Individual real-time PCR reactions for each virus were performed. After denaturing, PCR amplification was performed and profiles were collected and analyzed using Sequence Detection Software version 1.0. To quantify the three viruses, an external standard curve for each virus was established using 10-fold dilutions from 100 copies to $1.0 \times 10^6$ viral cDNA copies/PCR[52].

We employed similar methodology to quantify bacterial loads which were determined for stool samples positive for bacteria using singleplex real-time PCR assays for each respective bacteria (*Campylobacter, E. coli, Salmonella, Shigella*). Standard curves demonstrating the relationship between colony forming units (CFU) and crossing point of the real-time PCR assay for each organism were created by performing real-time PCR on 10-fold dilutions of standardized bacterial suspensions that were plated onto sheep blood agar plate to determine the CFU count.

**Statistical analysis.** For the primary outcome we a priori anticipated that follow-up would be complete for 90% of clinical trial participants ($N = 797$) and all of these participants would thus have follow-up MVS scores. Based on North American[19,53,54] data, we assumed the following pathogen distributions: ~50% viral ($N = 399$), ~40% unidentified ($N = 318$), and ~10% bacterial ($N = 80$). Given our 1:1 probiotic:placebo allocation ratio, we anticipated a minimum of 40 participants per arm in the smallest group[55]. The proposed minimum clinically important pathogen-group and pathogen-specific MVS difference of means were based on the natural history of disease[40], and the proposed benefits associated with probiotic administration (Supplementary Table 1). Power calculations assumed 40 subjects in each study arm pathogen group (i.e., probiotic and placebo virus, bacteria, and unidentified) and a standard deviation (SD) of 3.1[40] around the MVS score point estimates. Based on our proposed effect sizes and assuming a minimum of 40 paired specimens for each pathogen group comparison, power was greater than 80%. Similar power was present when the analysis was repeated with specific viral etiologies (smallest cell = 39).

For the primary outcome, clinical benefits were evaluated by comparing the mean post-randomization MVS scores between children who received probiotics versus placebo. The difference in means was explored in relation to pathogen-group (i.e., negative, virus only, bacteria only, virus/bacteria co-detection, and other co-detection including parasite only and parasite/virus co-detection), viral (i.e., rotavirus, norovirus, adenovirus), and bacterial agents (i.e., *Campylobacter* and *Salmonella* spp.). For co-detections (i.e., multiple enteropathogens detected), we first included all cases with the specific pathogen, then repeated the analysis using only single pathogen detection cases. Children under two years of age from whom only *Clostridioides difficile* was detected, were counted as test-negative as young children are often colonized with this agent[53,56]. To assess for main effects, the analysis employed a weighted linear regression model that included treatment, pathogen, interaction terms for treatment assignment and pathogen groups along with other key covariates (i.e., age, sex, MVS score at enrollment, pre-index visit or index visit hospitalization, and pre-index visit antibiotic use).

Stool pathogen load quantification values were log10 transformed in keeping with standards approaches to reporting nucleic acid concentrations in human body fluids. The secondary outcome of stool pathogen load reduction was quantified as the difference in the number of copies of pathogen-specific NA/gm between Days 0 and 5 (i.e., Day 0–Day 5 = stool pathogen load reduction) and Day 5 and 28. Multivariable regression models including treatment, pathogen and other key covariates (e.g., day of illness, age, sex, MVS score at enrollment, hospitalization, antibiotic use, baseline pathogen load) were constructed. Locally weighted smoothing lines were constructed and compared between groups.

Correlations between (1) the baseline (Day 0) stool pathogen load and the baseline MVS score and (2) the Day 5 stool pathogen load with the follow-up MVS score, were performed employing within pathogen-group and within pathogen (virus) linear regression analyses. All regression models were adjusted for a priori identified variables as described for the primary outcome.

All reported regression models employed variable transformations when model residuals were non-normally distributed and the transformations, if performed, are reported. Models were only constructed to evaluate pathogens identified in a minimum of 10 participants. Multiple imputation was used to account for individual missing elements of the 7-item MVS score. Time, but not date, of the first or last vomit or diarrheal episodes, were the most commonly missing variable, absent in a maximum of 24% of participants. The imputation model, based on inspection, assumed that data were missing at random and included key baseline characteristics, trial group, and all efficacy outcomes[14]. All statistical tests were two-sided; overall statistical tests of significance for the primary outcomes was set at 0.005 using the Bonferroni approach to correct for the 10 comparisons performed.

**Reporting summary.** Further information on research design is available in the Nature Research Reporting Summary linked to this article.

## Data availability

The full study protocol and the datasets, which includes all data fields reported in this study, are available, following manuscript publication, upon request from the corresponding author (Dr. Stephen Freedman, Stephen.Freedman@AlbertaHealthServices.ca), following the provision of ethics approval. The source data underlying Figs. 2 and 3a–c are provided as a Source Data file.

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

## Acknowledgements

We thank the participants and their families for trusting us to conduct this trial; the trial investigators and support staff across all sites for their commitment to the successful and diligent conduct of the trial; the site coordinators, including Marie-Christine Auclair, R.N., B.Sc.N., Zach Cantor, B.A., Kamary Coriolano Da Silva, Ph.D., Dale Dalgleish, B.H. Sc.N., Eleanor Fitzpatrick, M.N., Kelly Kim, B.Sc., B.A., Maggie Rumantir, M.D., Judy Sweeney, R.N., B.Sc.N., and Avery Yandt, B.Sc., B.A., for trial coordination; the staff of Lallemand Health Solutions, Calgary Laboratory Services, Alberta Health Services, Research Pharmacy, Provincial Laboratory for Public Health, Edmonton and Calgary, Alberta, for supporting the conduct of this trial; the emergency department physicians, nurses, and ancillary staff at all trial sites; the Pediatric Emergency Medicine Research Associates' Program at Alberta Children's Hospital, the Students Undertaking a Pediatric Program of Research Training program at the Children's Hospital of Eastern Ontario, and the staff of the Pediatric Research Academic Initiative in SickKids Emergency program at the Hospital for Sick Children, for identifying potentially eligible trial participants; and Dr. Marie Louie (Calgary) for building the connections that made the microbiologic investigations possible. No compensation for the assistance provided by any aforementioned persons was provided. This study was supported by the Canadian Institutes of Health Research (grants 286384 and 325412), the Alberta Children's Hospital Foundation Professorship in Child Health and Wellness (to Dr. Freedman), a grant from the Alberta Children's Hospital Foundation to the Pediatric Emergency Medicine Research Associates' Program, Calgary Laboratory Services (in kind), Provincial Laboratory for Public Health–Alberta Public Laboratories, Luminex, and Copan Italia. P.M.S. is the recipient of a Canada Research Chair in Gastrointestinal Disease.

## Author contributions

S.F. and Y.F. conceived the study, obtained funding, and S.F. wrote the first draft. S.F., X.P., L.C., B.L., P.S., D.S., and Y.F. designed the study with help from the other authors. S.W., S.S., S.G. K.F., N.P., and K.H. were responsible for clinical operations. S.W. oversaw all study activities. X.P., L.C., B.L. were in charge of developing microbiology protocols and standard operating procedures. Y.Q., B.G., C.L. was in charge of conducting all laboratory analyses and managed microbiology quality assurance. J.X. and A.N. conducted the data management and statistical analyses. S.W. ensured adherence to the protocol and handled data acquisition and institutional review board matters. All authors contributed in writing different sections of the manuscripts.

## Competing interests

None of the authors have any potential or perceived conflicts of interest to declare. The study sponsors played no role in study design, conduct, data acquisition, analysis, manuscript preparation or the decision to submit the manuscript for publication. Dr. Stephen Freedman provides consulting services to Takeda Pharmaceuticals Inc. and was an invited speaker at the 10th Probiotics, Prebiotics and New Foods, Nutraceuticals and Botanicals for Nutrition and Human and Microbiota Health. PMS serves on advisory boards for Antibe Therapeutics, Cargill and Nestle-Gerber. He has received honoraria from Abbott Nutrition for speaking at continuing medical education activities and received research support from Lallemand Human Nutrition for a postdoctoral fellow award through a Mitacs Accelerates Internship.

## Additional information

## the Pediatric Emergency Research Canada Probiotic (PERC) Regimen for Outpatient Gastroenteritis Utility of Treatment (PROGUT) Trial Group

Stephen B. Freedman[1], Jianling Xie[1], Alberto Nettel-Aguirre[1], Xiao-Li Pang[2], Linda Chui[2], Sarah Williamson-Urquhart[1], David Schnadower[3], Suzanne Schuh[4], Philip M. Sherman[4], Bonita Lee[2], Serge Gouin[5], Ken J. Farion[6], Naveen Poonai[7], Katrina F. Hurley[8], Yuanyuan Qiu[2], Binal Ghandi[2], Colin Lloyd[2], Yaron Finkelstein[4], Andrew R. Willan[4], Ron Goeree[9], David W. Johnson[1], Karen Black[10], Marc H. Gorelick[11]

[9]PATH Research Institute, St. Joseph's Healthcare Hamilton, 25 Main Street West, Suite 2000, Hamilton, ON L8P 1H1, Canada. [10]University of British Columbia, 4480 Oak St, Vancouver, BC V6H 3N1, Canada. [11]University of Minnesota, 2525 Chicago Ave, Minneapolis, MN 55404, USA.

