## [Peer Review File · Nature Communications]

Reviewers' Comments:

Reviewer #1:

Remarks to the Author:

Given that this study was conducted in Canada, there's a disproportionate focus on the epidemiology and etiology of diarrheal disease in the US.

Similarly, the intro focuses mainly on the kinetics of viral infection at the expense of bacterial pathogenesis. Would have liked more presentation/discussion of the reasons that probiotics might work to restore the gut diversity after bacterial infection, as opposed to viral.

While the main results have been reported already and the primary outcome is not a focus of the current paper, it could be made clearer that one of the large trials that reported null finding was in fact this one. As written, that's not apparent.

Overall the statistical methods are sound -- with the following minor points

- A comparison of characteristics of participants who did/did not submit a second and third stool should be presented
- Its stated that only 18% give all three specimens -- the implications of this potential bias should be discussed.
- Should also be acknowledged that the study was clearly underpowered to examine outcomes in bacterial or parasitic infections. To this end, consider changing the title and focus of abstract, since really only specific viral pathogens were studied.

Reviewer #2:

Remarks to the Author:

This is an elegant analysis of a previously published large-scale, multi-center, placebo-controlled trial of probiotics (*L. rhamnosus*/*L. helveticus*) for reduction of acute gastroenteritis (AGE) in children. As the original work did not find a benefit for probiotics in this context, the authors sought to determine whether the effects/benefits of probiotics are pathogen-specific. They provide sound reasoning to pursue such analysis, including the shifts in viral causative agents of AGE (from Rota to Noro), which might explain the discrepancy in findings between their more recent trial and older works, as well as differences in probiotics and pathogens mechanisms of action. Their analysis did not find a specific symptomatic benefit for probiotics when children were stratified according to the likely causative agent (bacterial / viral / mixed / parasite). Based on quantification of the leading viral causative agents of AGE in stool samples, did not find benefit for probiotics on reduction of viral loads compared to placebo. There are other factors that have recently been suggested to affect probiotics efficacy which the authors should address or at least discuss, but nonetheless, this work answers a specific important question – whether probiotics have pathogen-specific effect in the context of AGE, and likely answer is no (I would only be more cautious regarding bacterial AGE, see below).

- While the outcomes defined by the authors focus on pathogen-specific effects of probiotics, I wonder whether these data can be used to answer additional questions pertaining to probiotics efficacy or lack thereof. For example, personalized effects of probiotics are emerging as an important feature of this therapeutic, pertaining to factors such as age, diet, differences in colonization etc. This should be added to the discussion, and I would suggest the following analyses:
 1. A sub-group analysis based on age (and/ preterm birth status) can be useful here, especially as previous studies raised concerns regarding the use of probiotics in very-low birth weight infants.
 2. The authors have data regarding feeding (exclusive breastfeeding vs. mixed-feeding), this has been

suggested to affect probiotics efficacy, a sub-group analysis based on these diet differences can be illuminating.

3. Differences in probiotics colonization capacity, both during supplementation and in the long-term, have been suggested to affect efficacy. While stool samples might not be useful for predicting intestinal colonization during supplementation, the authors could nonetheless use the genetic material collected from the pre- and post-probiotics supplementation to quantify *L. rhamnosus* and *L. helveticus* in stool samples as an indication of whether there are colonization-dependent personalized effects for probiotics.

- As the authors were not able to analyze probiotics specific effects against bacteria, I would suggest to more cautiously phrase the conclusion in the discussion as the lack of effect can only be attributed to probiotics effect on viruses.
- Table 1, Supplementary table 2 - Please provide a statistical test to compare between the study groups at baseline.
- Line 50: remove "it".
- Please clarify acronyms (AGE).

Comment	Response
Reviewer #1	
Given that this study was conducted in Canada, there's a disproportionate focus on the epidemiology and etiology of diarrheal disease in the US.	We had included relevant US data to contextualize for the larger target journal audience, but as suggested by Reviewer #1, as this study was conducted in Canada we have included Canadian prevalence data as well. Introduction, Paragraph 1: “It is estimated that 0.57 acute gastroenteritis (AGE) episodes per person-year occur in Canada, amounting to nearly 19.5 million episodes annually,¹ while in the United States, over 48 million episodes occur each year.^{2,3}”
The intro focuses mainly on the kinetics of viral infection at the expense of bacterial pathogenesis. Would have liked more presentation/discussion of the reasons that probiotics might work to restore the gut diversity after bacterial infection, as opposed to viral.	We have focused the Introduction on viral kinetics as the study itself focuses on viral kinetics. We do not believe a focus in the Discussion on bacterial kinetics would be scientifically valid beyond speculation, as it has not been the focus of this clinical trial.
While the main results have been reported already and the primary outcome is not a focus of the current paper, it could be made clearer that one of the large trials that reported null finding was in fact this one. As written, that’s not apparent.	Thank you for pointing this out – we have added clarity and emphasis in the Introduction, second paragraph: “Although the administration of probiotic agents to children with AGE and diarrhea has been recommended by international clinical guidelines,¹¹⁻¹³ we recently conducted one of the largest randomized clinical trials to date, and found no benefits associated with probiotic administration. The trial, which included 886 children with AGE, reported that those who received a 5-day course of L. helveticus/L. rhamnosus did not have significantly different odds of experiencing moderate-to-severe AGE following randomization compared with those administered a placebo [OR: 1.06 (95% CI, 0.77 to 1.46)].¹⁴ These findings were supported by a simultaneously conducted trial in the United States, which evaluated a different probiotic product, containing L. rhamnosus GG.¹⁵” We have also clarified a sentence in the last paragraph of the Introduction: “To address this point, as an integral part of the aforementioned placebo-controlled, randomized, parallel-arm, clinical trial,¹⁴ we identified pathogens in collected stool specimens to enable an evaluation of the ability of an orally administered probiotic to reduce symptom severity at a pathogen-specific level. We also assessed the changes, from baseline, in viral loads in stool specimens at the end of the probiotic course (on day 5 of treatment) and 4 weeks after randomization (on day 28), relative to placebo.”

A comparison of characteristics of participants who did/did not submit a second and third stool should be presented.	This is an excellent point and we have added Supplementary Table 18 to fill these data. There were, however, no clinically significant differences between groups.
It is stated that only 18% gave all three specimens -- the implications of this potential bias should be discussed.	We have added to a sentence in our Limitations Section: “Our study has several limitations. Only 18% of participants provided specimens at all three study time points. Consequently, we had fewer participants than was anticipated and thus several secondary analyses were potentially underpowered. While participants who submitted all three specimens could have differed from those who did not, there is no reason to assume a related systematic bias. Indeed, Supplementary Table 18 demonstrates that the two groups of children were clinically similar.”
It should also be acknowledged that the study was clearly underpowered to examine outcomes in bacterial or parasitic infections.	We agree, and have added to a sentence in our Limitations Section: “However, because of the small number of participants with bacterial and parasitic infections, we were unable to conduct robust pathogen load analyses related to bacterial and parasitic clearance.”
To this end, consider changing the title and focus of abstract, since really only specific viral pathogens were studied.	Thank you for raising this concept – we have revised both accordingly.
Reviewer #2	
While the outcomes defined by the authors focus on pathogen-specific effects of probiotics, I wonder whether these data can be used to answer additional questions pertaining to probiotics efficacy or lack thereof. For example, personalized effects of probiotics are emerging as an important feature of this therapeutic, pertaining to factors such as age, diet, differences in colonization etc. This should be added to the discussion, and I would suggest the following analyses:  1) A sub-group analysis based on age (and/ preterm birth status) can be useful here, especially as previous studies raised concerns regarding the use of probiotics in very-low birth weight infants. 2) The authors have data regarding feeding (exclusive breastfeeding vs. mixed-feeding), this has been suggested to affect probiotics efficacy, a sub-group analysis based on these diet differences can be illuminating. 	We appreciate these important suggestions made by the Reviewer regarding further sub-group analyses, and have addressed them and added these to the results section as exploratory outcomes:  1) We conducted a sub-group analysis related to age stratified by pathogen sub-groups (but do not have prematurity data). The interaction between treatment assignment and age was not statistically significant in the linear regression model, which employed the MVS score as the independent variable, adjusted for other covariates (p. 11). We also present the results of these sub-group analyses in Supplementary Table 19. Overall, no sub-groups were statistically significant after adjustment for multiplicity. 2) We similarly conducted a sub-group analysis related to age stratified by pathogen sub-groups. The interaction between treatment assignment and breastfeeding status was not statistically significant in the linear regression

	model, which employed the MVS score as the independent variable, adjusted for other covariates (p. 11). We also present the results of these sub-group analyses in Supplementary Table 20 – no sub-groups were statistically significant after adjustment for multiplicity. Citation: “Sub-group analyses were conducted based on participant age and pathogen detected. In these models, the interaction between treatment assignment and age was not statistically significant when the MVS score was set as the independent variable and the models were adjusted for other relevant covariates; Supplementary Table 19. Similarly, we analyzed the primary outcome, MVS score, based on breast-feeding status and found no evidence of interaction with treatment assignment; Supplementary Table 20.”
Differences in probiotics colonization capacity, both during supplementation and in the long-term, have been suggested to affect efficacy. While stool samples might not be useful for predicting intestinal colonization during supplementation, the authors could nonetheless use the genetic material collected from the pre- and post-probiotics supplementation to quantify L. rhamnosus and L. helveticus in stool samples as an indication of whether there are colonization-dependent personalized effects for probiotics.	We thank the Reviewer for the comment, however, we do not have data on colonization-dependent personalized effects at this time. However, both in the clinical trial and in this specific viral analysis, we did not identify any parameters, covariates or sub-groups that were associated with improved outcomes among those allocated to the probiotic group.
As the authors were not able to analyze probiotics specific effects against bacteria, I would suggest to more cautiously phrase the conclusion in the discussion as the lack of effect can only be attributed to probiotics effect on viruses.	We appreciate this suggestion and have revised the conclusion of the Discussion accordingly: “In conclusion, we observed no beneficial virus-specific clinical effects associated with the administration of a 5-day course of a L. helveticus/L. rhamnosus combination probiotic, for children with AGE. Similarly, probiotic administration did not result in more rapid clearance of viral pathogens from stool specimens, compared with placebo. These findings strengthen the conclusion that in children who present to an ED with viral-induced AGE, twice-daily administration of a combined L. rhamnosus–L. helveticus probiotic does not reduce the severity of AGE, or expedite the clearance of viruses in

	stool.”
Table 1, Supplementary table 2 - Please provide a statistical test to compare between the study groups at baseline.	While most journals prefer not to publish Table 1 demographic p-values, as per Reviewer #2’s request we have added them, and will leave the decision whether to include these values at the Editor’s discretion.
Line 50: remove “it”.	Thank you – revised accordingly.
Please clarify acronyms (AGE).	An abbreviations table has been added.

Reviewers' Comments:

Reviewer #2:

Remarks to the Author:

In the revised manuscript, the authors have addressed all my suggestions, either analytically or through text revisions. I believe that, after addressing potential contributors to probiotic efficacy heterogeneity (age, diet), the authors observation of a null finding is even stronger. The authors have also made several modifications in the title, abstract and discussion that place less emphasize on the underpowered study of bacterial and parasitic pathogens, and clearly address the limitations of the study. This is an important work for both clinicians and researchers, and I support publication of the manuscript in its current form.